# Educational video combined with augmented clinical support to improve CPAP use in patients with obstructive sleep apnea-hypopnea syndrome: A randomized controlled trial protocol

Sameh Msaad[1,2]*, Dorra Abdelmouleh[1,2], Rim Khemakhem[1,2], Rahma Gargouri[1,2], Nesrine Kallel[1,2], Manel Maalej[1,3], Manel Turki[4], Imen Chaari[1,5], Narjes Abid[6,7], Asma Younes[8,9], Najla Bahloul[1,2], Samy Kammmoun[1,2]

1 Faculty of Medicine of Sfax, University of Sfax-Sfax, Tunisia, 2 Department of Respiratory and Sleep Medicine, University Hospital Hedi Chaker, Sfax, Tunisia, 3 Psychiatry C department, University Hospital Hedi Chaker, Sfax, Tunisia, 4 Faculty of Pharmacy of Monastir, Monastir, Tunisia, 5 Psychiatry B department, University Hospital Hedi Chaker, Sfax, Tunisia, 6 Faculty of Medicine of Tunis, University of Tunis-Al Manar-Tunis, Tunisia, 7 Department of Respiratory Medicine, University Hospital Taher Maamouri, Nabeul, Tunisia, 8 Faculty of Medicine of Sousse, University of Sousse-Sousse, Tunisia, 9 Department of Pneumonology, University Hospital Taher Sfar, Mahdia, Tunisia

* pneumo1972@gmail.com

## Abstract

### Background

Poor patient adherence to continuous positive airway pressure **(CPAP)** remains a common challenging issue and a major cause of treatment failure in patients with obstructive sleep apnea-hypopnea syndrome **(OSAHS)**. Our study aims to test the hypothesis that therapeutic patient education **(TPE)** combined with augmented clinical support **(ACS)** at CPAP initiation would improve CPAP adherence and treatment outcomes compared to usual care.

### Patients and methods

We will perform a prospective, randomized, controlled, parallel-group trial including 60 adult patients newly diagnosed with severe OSAHS. Each patient will be randomly assigned to either the TPE group or the usual care group and then scheduled to start CPAP therapy within 1–2 weeks after the nocturnal polygraph recording. For the TPE group, CPAP initiation will be performed at the hospital during a 3-hour educational session that will include 4 workshops entitled: "What's OSAHS", "CPAP machine: what is it? How does it work? And what does it serve for?", "How to use your CPAP and fit your mask" and "How to deal with CPAP side effects". The educational team will include a sleep disorders specialist, a sleep nurse, and a CPAP technician. As educational tools, we will use a short storytelling video in the local language, live demonstrations and a daily desk calendar with 60 removable sheets and 1 tip on

**Data availability statement:** The study was approved by the Ethics Committee of the Hedi Chaker University Hospital of Sfax (Tunisia) and complied with the Declaration of Helsinki and good clinical practice guidelines. All participants will be asked to provide written informed consent. All data collected during this study will be treated with strict confidentiality. No datasets were generated or analysed during the current study. All relevant data from this study will be made available upon study completion. Identified research data will be made publicly available when the study is completed and published.

**Funding:** L'Association des Pneumo-Allergologues du Centre et du Sud Tunisiens : APACS, MD Dorra Abdelmouleh.

**Competing interests:** The authors have declared that no competing interests exist.

CPAP therapy on each sheet. Patients assigned to the usual care group will undergo CPAP initiation at home under the guidance of a CPAP technician and will not receive additional educational support beyond the documents provided by the manufacturers. Our primary outcome is unadjusted CPAP adherence measured in hours/night at 1, 3, and 6 months after CPAP initiation. Our secondary outcome is functional status at the 6-month follow-up, which included snoring, nasal obstruction symptoms, subjective quality of life, fatigue, emotional status, cognitive function, insomnia and excessive daytime sleepiness **(EDS)**.

## Conclusion

We designed this original protocol by combining TPE with ACS. We hope that our findings will help us improve CPAP adherence among Tunisian patients with OSAHS.

## 1. Background

Obstructive sleep apnea-hypopnea syndrome **(OSAHS)** is probably the most common sleep-related disorder in Tunisia, with recent data from an online survey suggesting that nearly 10% of middle-aged Tunisian adults are at high risk for OSAHS [1]. Untreated OSA is associated with significant cardiovascular and metabolic morbidity, impaired emotional health, neurocognitive dysfunction, increased accident risk, and all-cause mortality. While there are various treatment methods for OSAHS, continuous positive airway pressure **(CPAP)** remains the first-line and gold-standard therapy for symptomatic and moderate-to-severe OSAHS patients. There is plenty of evidence to show that CPAP significantly improves several long-term clinically important outcomes including excessive daytime sleepiness **(EDS)**, sleep-related quality of life, motor vehicle accidents, and blood pressure [2]. Good patient adherence is the key to CPAP therapy effectiveness. As yet, there is no consensual definition of good CPAP adherence in terms of the threshold level for daily CPAP use required to achieve and maintain therapeutic effectiveness. The US Medicare policy for continued coverage of a CPAP device requires objective evidence of CPAP use for ≥4 hours (h) per night on ≥ 70% of nights within a consecutive thirty-day period anytime during the first 90 days of therapy [3]. However, it's becoming increasingly recognized that the optimal threshold for sufficient nightly CPAP use may depend upon the therapeutic outcome in question and the patient's self-perception of symptom severity. Moreover, a linear dose-response relationship was found between CPAP use and several outcomes with greater benefit with increased nightly duration of use > 7 h per night.

Despite the well-established efficacy of CPAP therapy, poor patient adherence remains a common challenging issue and a major cause of treatment failure. According to a 2004 quantitative review, the highest rate of low therapeutic adherence was associated with sleep disorders when compared to 17 other chronic conditions, such as human immunodeficiency virus (HIV) infection, cancer, cardiovascular disease, renal disease, and diabetes mellitus [4]. In a systemic review of 82 studies conducted over a twenty-year timeframe (1994–2015), the overall poor CPAP adherence rate

based on a 7-h/night sleep time was 34.1%, and there was no significant improvement in CPAP adherence over the time frame. More interestingly, the nationwide claims data lake for Sleep Apnea **(ALASKA)**, a large-scale analysis of the French national healthcare database including 480,000 patients treated with CPAP, reported that almost half of patients discontinue CPAP use at 3 years [5].

Poor adherence to CPAP may result from device-related issues (mask type, air leak, CPAP side effects) or the patient's clinical condition (socioeconomic status, comorbidities, psychosocial issues, endophenotype of the disease).

Over the last two decades, several interventions have been proven to increase CPAP adherence, including techno-logical advances in CPAP devices, augmented clinical support **(ACS)** (more frequent appointments, telephone calls, home visits), therapeutic patient education **(TPE)**, cognitive behavioral therapy and motivational enhancement. There is an increasing tendency for most clinicians to offer their patients a combination of ACS and TPE. According to the World Health Organization **(WHO)** definition, TPE helps patients acquire or maintain the skills they need to self-manage their project life with a chronic disease in the best possible way. To achieve this, candidates for CPAP therapy should clearly understand their disease. They also need to have accurate expectations of the benefits and harms of treatment as many of them mistakenly believe that CPAP therapy will definitively cure their disease, while others may feel hesitant or scared about wearing a mask to sleep. Moreover, patients must be motivated enough to continue their treatment at home without medical supervision.

Various educational techniques and tools can be used such as interactive presentations, roundtables, treatment demonstrations, role-playing, photos, flyers, posters and videos. Both patients and healthcare providers prefer video format rather than text or orally staff-delivered information. The video message is supposed to be more attractive and persuasive as it provides easy and personalized access to information through simplified explanations and visual demon-strations. In addition, it can be self-administered by the patient, which would have the advantage of freeing up the staff's time and preserving the patient's privacy [6].

There is currently little empirical evidence about the impact of TPE on CPAP adherence with conflicting research find-ings. Three clinical randomized trials **(CRT)** of motivational enhancement therapy **(MET)**/motivational interviewing initiated before at-home CPAP treatment reported significantly improved adherence of 1.5 h per night, 1.7 h per night, and 2.0 h per night compared with the usual care approach. In contrast, a more recent study found that educational videos have no significant impact on CPAP adherence when compared with usual care.

In this context, we will conduct this study to test the hypothesis that TPE combined with ACS at CPAP initiation would improve CPAP adherence and treatment outcomes compared to usual care.

## 2. Methods

### 2.1. Study design and setting

We will perform a prospective, randomized, controlled, parallel-group trial comparing educational video support with usual care (The unique identification number for the registry is PACTR202412813486845). The study was approved by the Ethics Committee of the Hedi Chaker University Hospital of Sfax-Tunisia (Reference CE 16/2024 September 10, 2024) and complied with the Declaration of Helsinki and good clinical practice guidelines. All participants will be asked to provide written informed consent.

### 2.2. Patients

All patients referred to the Department of Respiratory and Sleep Medicine at Hedi Chaker University Hospital of Sfax-Tunisia, for suspected OSAHS will be screened for eligibility between January 2025 and July 2025. Sixty patients will be included. The study will be terminated by December 31st, 2025.

Patients will be determined eligible if they are aged 18 years old and above and newly diagnosed with severe OSAS (obstructive apnea-hypopnea index **(OAHI)** ≥ 30 events/ h) as revealed by an overnight type III polysomnography **(PSG)**.

Patients who have previously received CPAP therapy will not be included in the study. Additional exclusion criteria are refusal to participate, illiteracy, pregnancy, major mental disorders, severe heart failure with systolic fraction ejection **(SEF)** < 45%, severe neuromuscular diseases and oxygen supplementation.

### 2.3. Collected data

**2.3.1. Sociodemographic data.** The following sociodemographic data will be asked: age, sex, civil status, educational level, employment status, engagement in night work, nature of communities (urban or rural), socioeconomic level, social and health insurance scheme as well as anthropometric characteristics.

**2.3.2. Health status.** We will collect data on lifestyle habits related to alcohol and tobacco consumption, dietary habits, internet use, analgesics/anxiolytics/hypnotics consumption, physical activity, sleep-wake habits, and sleep duration. We will also collect data on prevalent morbidities such as nasal allergy, asthma, chronic obstructive pulmonary disease **(COPD)**, cardiovascular and metabolic diseases as well as depression.

**2.3.3. Nocturnal polygraph recording.** We will use an overnight type III PSG (Nox A1, ResMed, Australia) to diagnose OSAHS. The recording will be performed at home and include monitoring of respiratory flow using a nasal cannula, snoring with a contact microphone attached to the neck, thoracoabdominal movements using respiratory inductance respiratory **(RIP)** belts, transcutaneous arterial oxygen saturation **($SpO_2$)** and heart rate by pulse oximetry, as well as body position and sleep-wake pattern using actimetry. Each recording will be reviewed automatically followed by manual scoring according to the 2012 American Association of Sleep Medicine **(AASM)** manual for the scoring of sleep and associated events. The OAHI, the 3% oxygen desaturation index **(ODI)**, and the percentage of recording time with $SpO_2$ levels below 90% **(T90%)** will be collected. OSAHS severity will be graded according to OAHI as follows: non-OSAHS < 5, mild OSAHS for $5 \le OAHI < 15$, moderate OSAHS for $15 \le OAHI < 30$, and severe OSAHS for $OAHI \ge 30$.

**2.3.4. Outcome measures.** Our prespecified primary outcome is CPAP adherence measured in h/night at 1, 3, and 6 months after starting CPAP therapy. Our secondary outcome is functional status at the 6-month follow-up, which includes snoring, nasal obstruction symptoms, subjective quality of life, fatigue, emotional status, cognitive function, insomnia and EDS.

#### 2.3.4.1. CPAP therapy

AirSense S10 (ResMed) will be used for all enrolled patients with a pressure range of 4 cm H2O to 20 cm H2O. Each patient will be sized for an appropriate nasal CPAP mask from ResMed. After 2 weeks of at-home auto-CPAP titration, a fixed CPAP prescription follows according to data downloaded from the secure digital **(SD)** card of the device. The 95th percentile **(p95)** of the pressure titrated by Auto-CPAP will be used as a reference pressure level for fixed-CPAP therapy.

Data from the CPAP SD card, including residual AHI; total and daily usage time and air leak, will be recorded using the ResScan software. Good adherence to CPAP at 3 months will be defined as a CPAP use for ≥4 h per night on ≥ 70% of nights within a consecutive thirty-day period anytime during the last 90 days of therapy. To assess patients' perception of CPAP therapy on the morning following the first night of CPAP titration, we will use the Sleep Disorders Center CPAP Questionnaire **(SDC CPAP questionnaire)**. This self-administered questionnaire includes 6 items scored on a 10-point Likert scale, with responses ranging from 1 (no difficulty/no discomfort/positive attitude toward CPAP) to 10 (significant difficulty/ significant discomfort/ negative attitude towards CPAP). We used the mean CPAP perception score to create 2 categories of patients: poor CPAP perception (scores > 16) and good CPAP perception (scores ≤ 16). We hypothesized that a poor CPAP perception score would independently predict poor CPAP adherence during the first 30 days of therapy. [7].

#### 2.3.4.2. Functional outcomes

➢ **Snoring**

Snoring will be assessed based on the question "Do you snore" with the following responses: "never," "rarely," "sometimes," "often," and "almost always." Based on these responses, 3 groups will be created: [1] "Never" or "Rarely," [2] "Sometimes," and [3] "Often or Almost Always".

➢ **Nasal obstruction**

Chronic nasal obstruction is a common symptom among patients with OSAHS and is a known risk factor for OSAHS as well as CPAP intolerance and discontinuation. We will use a validated Arabic version of the Nasal Obstruction Symptom Evaluation **(NOSE)** scale to assess the impact of nasal obstruction on patients' quality of life [8]. This scale is a reliable and valid self-report questionnaire comprising 5 items related to nasal obstruction symptoms experienced over the past month. Each item is scored on a 5-point Likert scale, where 0 indicates "not a problem" and 3 indicates a severe problem. The sum of the 5-item-answers is multiplied by 5, which provides a total score ranging from 0 to 100 with higher scores indicating worse symptoms. NOSE scores of [5–25], [26–50], and [50–100] represent mild, moderate, and severe nasal obstruction-related symptoms, respectively.

➢ **Subjective quality of life**

We will use a validated Arabic version of the World Health Organization- Five Well-Being Index **(WHO-5)** to evaluate the quality of life [9]. The WHO-5 is a concise self-assessment questionnaire designed to gauge subjective well-being over the previous two weeks. It comprises five straightforward, positively phrased items that evaluate positive mood (good spirits, relaxation), vitality (being active and waking up fresh and rested), and general interest (being interested in things). Each item is graded on a 6-point Likert scale, ranging from 0 (at no time) to 5 (all the time). The total raw score is calculated by summing the scores of the 5 individual items and ranges from 0 to 5. The final score is obtained by multiplying the total raw score by 4, resulting in a total score ranging from 0 to 100, with higher scores indicating better well-being. A final score of 50 or lower suggests poor well-being.

➢ **Fatigue**

We will use an Arabic-validated version of the fatigue severity scale **(FSS)** to measure fatigue in our participants [10]. The FSS is a self-report questionnaire with 9 items that assess the severity, impact, and frequency of fatigue on different aspects of daily life over the past week. Each item is rated on a scale from 1 to 7, where 1 indicates strong disagreement with the statement and 7 indicates strong agreement with the statement. This results in a total score ranging from 9 to 63. A total score of 36 or higher suggests moderate to severe fatigue.

➢ **Emotional status**

We will use an Arabic-validated version of the Hospital Anxiety and Depression Scale **(HADS)** to assess anxiety and depression in our patients [11]. The HADS is a 14-item questionnaire specifically developed to gauge anxiety and depression in non-psychiatric populations. It comprises two subscales, one for depression and the other for anxiety, each consisting of 7 items. Based on the relative frequency of symptoms over the past week, the respondent rates each item on a 4-point Likert scale ranging from 0 (no impairment) to 3 (severe impairment). The total score is derived by summing responses of all 14 items resulting in a total score ranging from 0 to 21 for each subscale. Scores of 11 or more on either subscale indicate significant symptoms, while scores of 8–10 indicate 'borderline' and 0–7 indicate 'normal'.

➢ **Cognitive function**

We will use a validated Arabic version of the Montreal Cognitive Assessment **(MoCA)** to evaluate the cognitive status of our patients [12]. The MoCA is a widely used tool for early screening for mild cognitive impairment **(MCI)** in the general population. It assesses various cognitive domains, including orientation, memory, attention, language, executive function, and visuospatial function. These domains are evaluated using tasks such as naming three uncommon animals, recalling five words, performing a simple Trail Making Test Part B, and copying a cube and other items. The total MoCA score, obtained by adding points from each completed task, ranges from 0 to 30 points, with higher scores indicating better cognitive functioning. Patients with a score of 25 or lower are suspected of having cognitive impairment.

➢ **Insomnia**

Insomnia will be evaluated using a validated Arabic version of the Insomnia Severity Index **(ISI)** [13]. The ISI consists of seven self-report items aimed at assessing various aspects of insomnia, including the severity of sleep onset, maintenance, and early morning awakening problems, as well as sleep dissatisfaction, interference with daytime functioning,

noticeability of sleep problems by others, and distress related to sleep difficulties. Each item is rated on a 5-point Likert scale resulting in a total score of 0–28, with higher scores indicating more severe insomnia. Participants are classified as having a high risk of insomnia if their ISI score is 15 or higher.

➢ **Excessive daytime sleepiness**

The study will use a validated Arabic version of the Epworth Sleepiness Scale **(ESS)** to measure daytime sleepiness over the past 3 months [14]. The ESS is an 8-item questionnaire where participants rate their likelihood of dozing off or falling asleep in various everyday situations on a 4-point Likert scale of 0–3. The total ESS score, calculated by summing all 8 item scores, ranges from 0 to 24, with higher scores indicating more severe daytime sleepiness. In this study, EDS is defined as an ESS score of 11 or higher, while severe EDS is defined as an ESS score of 16 or higher.

## 2.4. Therapeutic patient education

**2.4.1. Strategy.** To achieve our main objective, which is optimal adherence to CPAP, our TPE strategy focuses on explaining to the patient his disease and the benefit of being treated, leaving the principle that the better the patient understands, the better he will adhere to his treatment. In addition, the educational team will be provided with a detailed schedule for each TPE session indicating the order and timing of different workshops. The idea is to provide all patients with a common TPE program through a patient-centred approach.

The different TPE workshops were built through meticulous preparation by the educational team. Workshop topics were determined based on the identified obstacles to CPAP use. These obstacles included mainly the non-understanding of the diseases and the benefit of being treated, chronicity-dependence (trouble getting used to wearing the CPAP mask), having trouble properly fitting his CPAP mask, and side effects of CPAP.

**2.4.2. Team.** The educational team will include a sleep disorders specialist, a sleep nurse, and a CPAP technician.

**2.4.3. Tools.** For TPE at the hospital, we will use a short educational video (duration 10 minutes (min)) presenting a wide range of medical information in realistic clinical scenarios. The video script was written in Arabic Tunisian dialect through storytelling techniques to simplify medical jargon and help patients in fully comprehend their disease and the treatment they will receive (Video 1).

The video aims to provide patients with clear information about the OSAHS mechanism, risk factors, symptoms and complications. We will also use a before-after scenario to show patients the benefits of CPAP adherence by highlighting before and after CPAP effects. The video also offers tips to help patients adjust to using a CPAP machine and shows how using a CPAP machine will get easier over time. Additionally, the video explains the common side effects of CPAP and what patients can do to manage them.

To follow up on the TPE at home, every patient will be provided with a daily desk calendar and 1 tip on CPAP therapy on each sheet (Video 2). This strategy of daily reminders through brief sayings or phrases will serve as a proactive communication tool to keep patients informed and engaged in their healthcare journey.

**2.4.4. Education location.** During TPE sessions, patients will gather in a spacious, calm room with light and airy wall colours: white and cream. The room is furnished with a flat-screen high-definition television, a board, flexible chairs, and wireless internet. The chairs are arranged in a U-shape so that patients can face each other, while the health provider can move around the room.

**2.4.5. Educational session program.** As shown in Table 1, the TPE session will start at 9:00 AM and usually last for about 3 hours. Patients are expected to bring their CPAP devices with them. Upon arrival, they will be greeted by the sleep nurse, after which the day's agenda will be presented and the therapeutic education team introduced. Patients will be then encouraged to share their expectations for the session.

The TPE session program will include 4 workshops entitled: "What's OSAHS", "CPAP machine: what is it? How does it work? And what does it serve for?", "How to use your CPAP and fit your mask" and "How to deal with CPAP side effects".

**Table 1. TPE session schedule at the hospital CPAP initiation.**

| |
|---|
| **09:00 am** Reception of patients |
| **09:10 am** Introduction of therapeutic educational teal and day's agenda to patients |
| **09:20 am** Video-projection |
| **09:50 am** W1:" What's OSAHS" |
| **10:05 am** W2: "CPAP machine: what is it? How does it work? And what does it serve for?", Each patient disassembles, cleans, and reassembles his CPAP device. |
| **10:35 am** W3: "How to use your CPAP and fit your mask" Each patient wears his CPAP mask for at least 20 min. |
| **11:05 am** W4: "How to deal with CPAP side effects" |
| **12:00 pm** Return home with CPAP device and educational support. Each patient is provided with educational support (desk Calander and educational documents provided by the manufacturer). |
| TPE: therapeutic patient education. CPAP: continuous positive airway pr" pressure. W: workshop. |

All workshops will combine video demonstrations with live presentations. As a result, each patient will be invited to disassemble, clean, and reassemble his CPAP device. Patients will be then acclimated to wear the CPAP mask for at least 20 min during the TPE session. All these activities will be supervised and guided by a CPAP technician.

## 2.5. Usual care

Patients assigned to the usual care group will undergo CPAP initiation at home under the guidance of a trained CPAP technician. The initiation session will last 30–40 min and include an oral explanation of OSAHS and CPAP therapy and a brief demonstration of CPAP device usage. These patients will not receive additional educational support beyond the documents provided by the manufacturers and will have no direct contact with investigators before the 1-month follow-up appointment.

## 2.6. Protocol study

After a nocturnal polygraph recording showed severe OSAHS, the patient will be seen by a sleep doctor the following day. The doctor will explain the diagnosis and the need for CPAP therapy. The patient will then complete an informed consent form to participate in the study and be randomly assigned to either the usual care (n = 30) or the TPE group (n = 30). Additionally, the patient will fill out a questionnaire covering sociodemographic data, health status, NOSE questionnaire, WHO-5 questionnaire, HAD questionnaire, FSS questionnaire, MoCA questionnaire, ESS questionnaire and ISI questionnaire. Following this, the patient will be scheduled to start CPAP therapy within 1–2 weeks after the nocturnal polygraph recording.

Following the TPE session, patients in the TPE group will be provided with CPAP devices to take home. They will be advised to call the sleep nurses if they experience any side effects. The phone line will be available on weekdays from 9:00 am to 4:00 pm, and patients can leave a message at other times. An assigned nurse will contact patients on the second and fifteenth days to identify and resolve any CPAP-related issues. The SDC CPAP Questionnaire will be administered by the nurse for each patient in this group on day 2 following CPAP titration.

Patients in the usual care group will be advised to consult the CPAP technician for any questions or concerns.

Both groups of patients will be scheduled for follow-up appointments with a CPAP technician and a sleep physician on day 15 (only if the CPAP perception score is poor) and at 1, 3, and 6 months after starting CPAP therapy. During each visit, we will assess adherence to and tolerance of CPAP by asking questions about side effects and reviewing CPAP SD Card data. Any CPAP-related issues reported by the patients will be addressed accordingly. At the 6-month follow-up, all patients will be asked to complete the functional outcomes questionnaires once again (Figs 1 and 2). Sleep physicians involved in the routine clinical follow-up of the patients will be blinded to each patient's treatment group.

| | Study period | | | | | | | | | | | | | | | |
|---|---|---|---|---|---|---|---|---|---|---|---|---|---|---|---|---|
| | Enrollment | | CPAP initiation | | Post CPAP initiation | | | | | | | | | | Close-out | |
| Timepoint (day ± window) | -7 to15* | | Day1 | | Day 2 | | Day15 | | M1 | | M3 | | M6 | | December 31, 2025 | |
| | G1 | G2 | G1 | G2 | G1 | G2 | G1 | G2 | G1 | G2 | G1 | G2 | G1 | G2 | G1 | G2 |
| Inclusion exclusion criteria | X | | | | | | | | | | | | | | | |
| Informed consent | X | | | | | | | | | | | | | | | |
| Sociodemographic data and health status | X | | | | | | | | | | | | | | | |
| NOSE, WHO5, FSS, HADS, MoCA, ISS and ESS | X | | | | | | | | | | | | X | | X | |
| Auto-CPAP initiation | | | X | | | | | | | | | | | | | |
| Switch to fixed CPAP | | | | | | | X | | | | | | | | | |
| Usual care | | | | X | | | | | | | | | | | | |
| TPE | | | X | | | | | | | | | | | | | |
| Call Phone | | | | | X | | X | | | | | | | | | |
| SDC CPAP Questionnaire | | | | | X | | | | | | | | | | | |
| CPAP side effects | | | | | X | | X | | X | | X | | X | | X | |
| SD Card data | | | | | | | X** | | X | | X | | X | | X | |

CPAP: Continuous Positive Airway Pressure. M: Month. G1: Group with therapeutic patient education. G2: Group with usual care. NOSE: Nasal Obstruction Symptom Evaluation. WHO5: World Health Organization- Five Well-Being Index. FSS: Fatigue Severity Scale. HADS: Hospital Anxiety and Depression Scale. MoCA: Montreal cognitive assessment. ISS: Insomnia Severity Index. ESS: Epworth Sleepiness Scale. TPE: Therapeutic Patient Education. SDC CPAP Questionnaire: Sleep Disorders Center CPAP Questionnaire. *: Enrollment is performed 1 to 2 weeks before CPAP initiation. **: only if poor CPAP perception score.

**Fig 1. SPIRIT schedule of enrollment.**

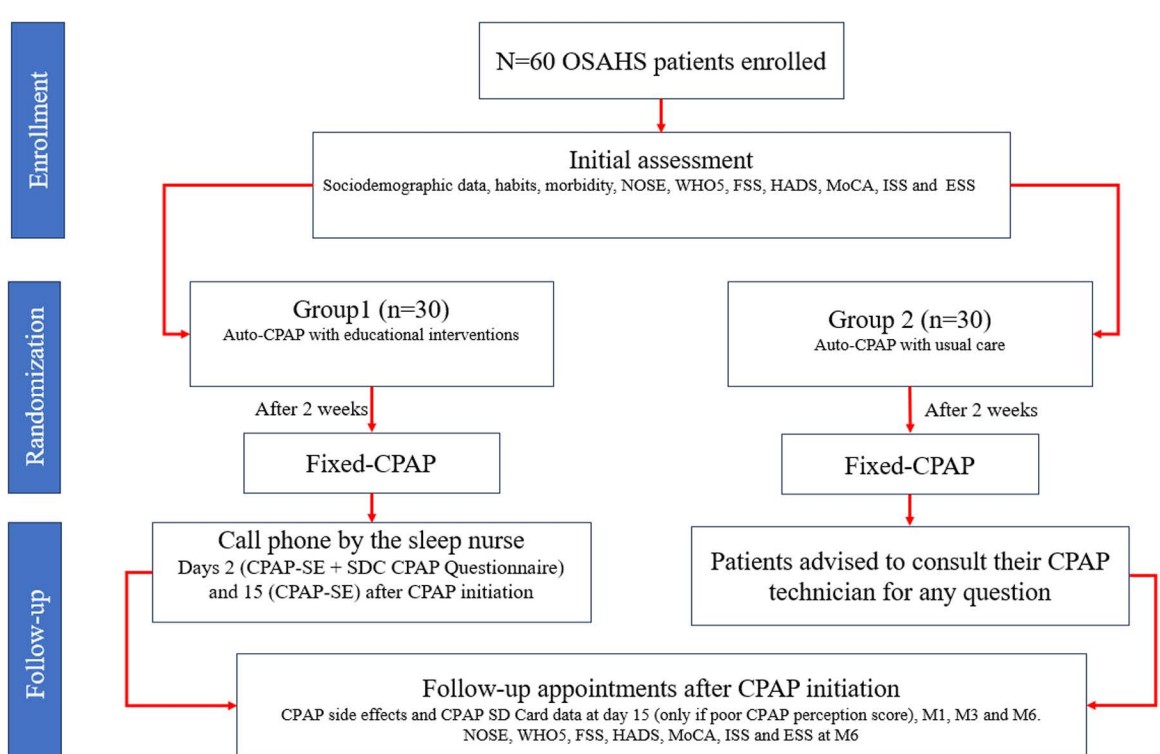

**Fig 2. Flow diagram of the study protocol.** OSAHS: obstructive sleep apnea-hypopnea syndrome. NOSE: Nasal Obstruction Symptom Evaluation. WHO5: World Health Organization- Five Well-Being Index. FSS: Fatigue Severity Scale. HADS: Hospital Anxiety and Depression Scale. MoCA: Montreal cognitive assessment. ISS: Insomnia Severity Index. ESS: Epworth Sleepiness Scale. CPAP: Continuous Positive Airway Pressure. SE: Side Effects. SDC CPAP Questionnaire: Sleep Disorders Center CPAP Questionnaire. M: Month.

## 2.7. Statistical analysis

The data analysis will be conducted using SPSS Statistics for Windows, Version 23.0 (IBM SPSS Statistics for Windows, Version 23.0. Armonk, NY: IBM). We will use the Kolmogorov-Smirnov test to assess the normal distribution of data. Continuous variables with a normal distribution will be expressed as mean and standard deviation **(SD)**, while non-normally distributed variables will be represented as median and interquartile range **(IQR)**. For comparing means of 2 and 3 or more categories of normally distributed continuous variables, independent Samples t-test and ANOVA test will be used, respectively. Medians of non-normally distributed continuous variables will be compared using non-parametric tests (Mann-Whitney U test and Kruskal-Wallis). Categorical variables will be presented as counts and frequencies. The frequencies of categorical variables will be compared using Pearson χ2 or Fisher's exact test, as appropriate. A significant level of $p < 0.05$ will be considered.

# 3. Discussion

## 3.1. Strength of the study

To the best of our knowledge, this study is the first RCT in Tunisia and the whole of Africa to assess the impact of TPE on adherence and effectiveness of CPAP therapy in patients with OSAHS. In Tunisia, there is no updated data about CPAP therapy. However, it's admitted that there is a rapidly growing number of patients treated with CPAP and almost all of them are provided with usual care. Most sleep specialists in Tunisia complain about the high rate of poor CPAP adherence among their patients. Therefore, it's more than needed to develop an appropriate approach to improve adherence and effectiveness of CPAP Therapy. In our study, we designed a pragmatic RCT that can be used in real-life routine practice conditions. The originality of this RCT is that we combine TPE via educational video with ACS by phone call. The content of TPE sessions as well as the phone discussion between the sleep specialists and the patients were standardized so that all patients would receive almost the same messages. We also develop a short storytelling video in the local language to be more captive and better connect with patients. To reduce bias, we decided to assess adherence and effectiveness of CPAP using objective CPAP SD card data. We will also use a range of validated self-reported measures in the local language for a more accurate assessment of subjective functional outcomes.

The study results will be shared with different healthcare providers to encourage them to change their approaches to their patients. The changes will improve patient care and provider efficiency. The study team believes that improving CPAP adherence in patients with OSAHOS should be a priority, and they hope that their results will help achieve this objective.

## 3.2. Limitations of the study

There are a few limitations that we need to consider in this study. Firstly, our cohort consists of a small sample treated in a tertiary referral center and therefore our findings may not be generalizable for other levels of care. Despite efforts made to ensure that patient care will be protocol-driven, some inadvertent biases would not be eliminated. For example, the sleep specialists involved in the routine clinical follow-up of the patients were blinded to each patient's study group, but the sleep nurses and CPAP technicians supporting the patients were not. Additionally, the content of educational tools and phone conversations was standardized. However, messages provided to patients may not be uniform because of the disparities in communication skills between education providers. We cannot also exclude the eventuality that some of our patients assigned to usual care accessed online educational materials outside of the context of the clinical trial.

Patients in this trial will be followed up for only 6 months, whereas CPAP is usually envisaged as a lifelong treatment. Thus, large-scale RCTs with long-term follow-up are required to determine whether the benefit of TPE combined with ACS on CPAP adherence and effectiveness is sustained or tends to be attenuated over time.

## 4. Conclusion

Poor patient adherence to CPAP remains a common challenging issue and a major cause of treatment failure. Previous studies on the impact of TPE on CPAP adherence have led to conflicting findings. In this context, we decided to conduct a prospective, randomized, controlled, parallel-group trial to assess the impact of TPE combined with ACS on CPAP adherence and effectiveness when compared to usual care. Our educational materials include a short storytelling video in the local language, live demonstrations, a daily desk calendar with 60 removable sheets and 1 tip on CPAP therapy on each sheet, and educational support provided by the manufacturers. Patients will be followed up for 6 months. Our primary outcome is unadjusted CPAP use measured in hours/night, and our secondary outcome is OSAHS-related functional outcomes. We firmly believe that improving CPAP adherence should be considered a priority in patients with OSAHOS, so to achieve this goal, we designed this original protocol combining TPE with ACS.

## Supporting information

**S1 File. The CONSORT 2010 checklist.**
(DOCX)

**S2 File. The research project presentation sheet.**
(PDF)

**S3 File.** Video 1: Be khayr – بخير. (Dec 15, 2024). إعلان فيلم ليليتك زينة – Offical trailer Liltek Zina I. https://www.youtube.com/watch?v=qzzCzxLSJts. Video 2: Be khayr – بخير. (Apr 3, 2025). AGENDA_SAS. https://www.youtube.com/watch?v=iirm-0Oi6YU&t=1s.
(DOCX)

## Author contributions

**Conceptualization:** Msaad Sameh, Dorra Abdelmouleh, Nesrine Kallel, Manel Maalej, Manel Turki, Imen Chaari, Samy Kammmoun.

**Data curation:** Nesrine Kallel.

**Investigation:** Msaad Sameh, Dorra Abdelmouleh, Rahma Gargouri, Nesrine Kallel, Narjes Abid, Asma Younes.

**Methodology:** Msaad Sameh, Dorra Abdelmouleh, Rahma Gargouri, Nesrine Kallel, Manel Turki, Imen Chaari.

**Project administration:** Manel Turki, Samy Kammmoun.

**Supervision:** Najla Bahloul, Samy Kammmoun.

**Validation:** Manel Maalej, Imen Chaari, Samy Kammmoun.

**Visualization:** Rahma Gargouri.

**Writing – original draft:** Msaad Sameh.

**Writing – review & editing:** Rim Khemakhem, Imen Chaari.

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
