## [Decision Letter · Decision Letter 0]

6 Feb 2025

PONE-D-24-46553Educational video combined with augmented clinical support to improve CPAP use in patients with obstructive sleep apnea-hypopnea syndrome: a randomized controlled trialPLOS ONE

Dear Dr. Sameh,

Thank you for submitting your manuscript to PLOS ONE. After careful consideration, we feel that it has merit but does not fully meet PLOS ONE’s publication criteria as it currently stands. Therefore, we invite you to submit a revised version of the manuscript that addresses the points raised during the review process.

We look forward to receiving your revised manuscript.

Kind regards,

Yongzhong Guo, Ph.D

Academic Editor

PLOS ONE

2. In the Methods section of your revised manuscript, please include the full name of the institutional review board or ethics committee that approved the protocol, the approval or permit number that was issued, and the date that approval was granted.

3. Please include “Protocol” in the manuscript title.

Reviewers' comments:

Reviewer's Responses to Questions

**Comments to the Author**

1. Does the manuscript provide a valid rationale for the proposed study, with clearly identified and justified research questions?

Reviewer #1: Yes

Reviewer #2: Yes

Reviewer #3: Partly

2. Is the protocol technically sound and planned in a manner that will lead to a meaningful outcome and allow testing the stated hypotheses?

Reviewer #1: Yes

Reviewer #2: Yes

Reviewer #3: Partly

3. Is the methodology feasible and described in sufficient detail to allow the work to be replicable?

Reviewer #1: Yes

Reviewer #2: Yes

Reviewer #3: Yes

4. Have the authors described where all data underlying the findings will be made available when the study is complete?

Reviewer #1: No

Reviewer #2: Yes

Reviewer #3: No

5. Is the manuscript presented in an intelligible fashion and written in standard English?

Reviewer #1: Yes

Reviewer #2: Yes

Reviewer #3: Yes

6. Review Comments to the Author

You may also provide optional suggestions and comments to authors that they might find helpful in planning their study.

Reviewer #1: Below are my comments:

- Please number the references and list them after the conclusion in the order they appear in the manuscript.

- Please mention the entire process including data collection and intervention in the past tense.

- A 1 week data might be too little for changing auto CPAP to fixed CPAP. Have compliance, residual AHIs, leaks, etc accounted for before switching to fixed CPAP?

- While nasal obstruction is a common symptom and a risk factor, snoring is even more commonly present in OSA. Has snoring been assessed?

- While the methods have been described, I do not find the results of the study anywhere in the manuscript.

- The manuscript cannot be assessed without complete data.

Reviewer #2: Overall, the study protocol looks good. It would be great to have real data. It may bring more meaningful outcome to this important field.

Reviewer #3: Here is a list of specific comments. Note: line numbering is not available, and page numbering in reviews

and comments is based on ruler applied in Editorial Manager-generated PDF.

1. I did not find the CONSORT checklist.

2. I did not find the study protocol as a supplement.

3. Page 6, Section 2.3.4, line 1: I suggest removing “unadjusted.” Adjusting or not would be part of

statistical analysis plan, and does not affect the definition of the outcome.

4. Page 6, Section 2.3.4.1, lines 8–10: Should “good adherence” at 3 months be one of the outcomes?

5. Page 6, Section 2.3.4.1, lines 11–13: Should “patients’ perception of CPAP” be one of the outcomes?

It might not be any difference between groups because the questionnaire was administered after the

first night of CPAP tiration. Was the questionnaire administered again later (e.g., 1, 3, or 6 months)?

What was the reason for administering the questionnaire? Was the questionnaire used to tailor the

EPT workshops?

6. Page 8, Section 2.4.1, line 1: It was not clear how the statistical analysis in Section 2.7 can answer

“optimal adherence to CPAP.”

7. Page 8, Section 2.4.1, line 5: I suggest spelling out EPT at its first appearance.

8. Page 9, Section 2.4.2, lines 1–2: The sentence appeared twice within few lines on this page. I suggest

reducing the redundant words as much as possible.

9. Page 9, Section 2.4.5: I suggest calling out Table 1 in Section 2.4.5. The previous calling out in

Section 2.4.1 may not be ideal.

10. Page 10, Section 2.6, lines 21–22: This answered Comment 5 above. I suggest revising the earlier

sentence where Comment 5 referred to.

11. Page 11, Section 2.7: Were there sample size and power calculation to determine the required sample

sizes per group?

12. Page 11, Section 2.7, line 7: The intervention had two groups. I suggest clarifying what “3 or more

groups” referred to.

13. Table 1: There were only two timestamps, reception and return. I suggest adding a timestamp for each

row. Also, please check the return home timestamp. Was 12 AM correct?

7. PLOS authors have the option to publish the peer review history of their article (what does this mean? ). If published, this will include your full peer review and any attached files.

**Do you want your identity to be public for this peer review?** For information about this choice, including consent withdrawal, please see our Privacy Policy .

Reviewer #1: No

Reviewer #2: No

Reviewer #3: No

---

## [Author Response · Author response to Decision Letter 1]

16 Feb 2025

February 16, 2025

Manuscript-Resubmission: [Educational video combined with augmented clinical support to improve CPAP use in patients with obstructive sleep apnea-hypopnea syndrome: a randomized controlled trial Protocol].

Manuscript Number: PONE-D-24-46553 

Academic Editor PLOS ONE: Yongzhong Guo, Ph.D

Dear Editor and Reviewers,

Thank you for your insightful comments. On behalf of my co-authors, I sincerely appreciate the time and effort you have dedicated to reviewing our manuscript titled "Educational Video Combined with Augmented Clinical Support to Improve CPAP Use in Patients with Obstructive Sleep Apnea-Hypopnea Syndrome: A Randomized Controlled Trial Protocol."

Your feedback has been invaluable. We have carefully considered your suggestions and revised the manuscript accordingly. Our responses to your comments are provided in a point-by-point format below. Additionally, we have submitted a revised version of the manuscript, with changes highlighted in yellow.

We look forward to your response regarding our revisions and are more than willing to make any further changes that would enhance the paper. Thank you once again for your attention and consideration.

Best regards,

Sameh MSAAD, MD

Faculty of Medicine. University of Sfax-Tunisia

Journal requirements

Response: As you recommended, we have carefully checked that our manuscript meets PLOSE ONE’s requirements. References were presented in Vancouver style.

2. In the Methods section of your revised manuscript, please include the full name of the institutional review board or ethics committee that approved the protocol, the approval or permit number that was issued, and the date that approval was granted.

Response: We apologize for this oversight. All necessary information regarding ethical approval was included in the revised manuscript.

3. Please include “Protocol” in the manuscript title.

Response: As you suggested, we included “Protocol” in the manuscript title.

4. We note that the grant information you provided in the ‘Funding Information’ and ‘Financial Disclosure’ sections do not match. When you resubmit, please ensure that you provide the correct grant numbers for the awards you received for your study in the ‘Funding Information’ section.

Response: We sincerely apologize for this oversight. We have thoroughly reviewed and rectified the information regarding the funding of our protocol.

Reviewers' comments

Reviewer #1

1. Please number the references and list them after the conclusion in the order they appear in the manuscript.

Response: Thank you for your comment. Following your suggestion, references have been organized according to the Vancouver style as outlined in the guidelines of PLOS ONE.

2. Please mention the entire process including data collection and intervention in the past tense.

Response: At the time of protocol drafting, data collection and intervention had not yet begun. Therefore, we describe the entire process in the future tense.

3. A 1-week data might be too little for changing auto CPAP to fixed CPAP. Have compliance, residual AHIs, leaks, etc accounted for before switching to fixed CPAP?

Response: Response: We agree with this assessment; therefore, we have decided to extend the Auto-CPAP titration duration to two weeks instead of one.

4. While nasal obstruction is a common symptom and a risk factor, snoring is even more commonly present in OSA. Has snoring been assessed?

Response: Thank you for your interesting suggestion. We have added a section for assessing snoring. Snoring will be evaluated using the question, "Do you snore?" with the following response options: "never," "rarely," "sometimes," "often," and "almost always." Based on these responses, participants will be categorized into three groups: (1) "Never" or "Rarely," (2) "Sometimes," and (3) "Often" or "Almost Always."

5. While the methods have been described, I do not find the results of the study anywhere in the manuscript.

Response: At the time the protocol was drafted, data collection and intervention had not yet started. Therefore, we do not have any results at this point.

6. The manuscript cannot be assessed without complete data.

Response: This is a protocol study scheduled to take place from January 2025 to July 2025, with a planned conclusion by December 31, 2025. As a result, we do not have any results available at this time. We look forward to sharing our findings with you in our upcoming paper.

Reviewer #2

Overall, the study protocol looks good. It would be great to have real data. It may bring more meaningful outcomes to this important field.

Response: Thank you for your valuable comment and suggestion. We would like to inform you that this protocol study is scheduled to occur from January 2025 to July 2025, with a planned conclusion by December 31, 2025. Consequently, we do not have any results to share at this moment. We look forward to presenting our findings in our forthcoming paper.

Reviewer #3

Here is a list of specific comments.

Note: line numbering is not available, and page numbering in reviews

and comments is based on ruler applied in Editorial Manager-generated PDF.

1. I did not find the CONSORT checklist.

Response: Thank you for your remark. the CONSORT checklist has been already submitted to the journal.

2. I did not find the study protocol as a supplement.

Response: Thank you for your remark. The study protocol as a supplement has been already submitted to the journal.

3. Page 6, Section 2.3.4, line 1: I suggest removing “unadjusted.” Adjusting or not would be part of the statistical analysis plan, and does not affect the definition of the outcome.

Response: We appreciate your input and have followed your recommendation by removing the term “unadjusted” in Section 2.3.4.

4. Page 6, Section 2.3.4.1, lines 8–10: Should “good adherence” at 3 months be one of the outcomes?

Response: We consider "good adherence" at 3 months to be a primary outcome of our study. This data is mentioned more clearly in the manuscript.

5. Page 6, Section 2.3.4.1, lines 11–13: Should “patients’ perception of CPAP” be one of the outcomes? It might not be any difference between groups because the questionnaire was administered after the

the first night of CPAP titration. Was the questionnaire administered again later (e.g., 1, 3, or 6 months)?

What was the reason for administering the questionnaire? Was the questionnaire used to tailor the

EPT workshops?

Response: Thank you for highlighting the need for clearer details. The Sleep Disorders Center CPAP Questionnaire (SDC CPAP Questionnaire) will be administered only once, on the morning following the first night of CPAP titration. The responses to this questionnaire can predict how well a patient will adhere to CPAP therapy during the first 30 days. This information allows clinicians to identify patients at risk of non-adherence to treatment. The SDC CPAP Questionnaire will be administered by the nurse for each patient in the PET group on day 2 following CPAP initiation. We will use the mean CPAP perception score to create 2 categories of patients: poor CPAP perception (scores > 16) and good CPAP perception (scores ≤ 16). We hypothesize that a poor CPAP perception score would independently predict poor CPAP adherence during the first 30 days of therapy) (Balachandran et al., 2013). For patients in the TPE group with a poor CPAP perception score, a supplementary follow-up appointment with a sleep physician will be scheduled for day 15 after CPAP initiation.

6. Page 8, Section 2.4.1, line 1: It was not clear how the statistical analysis in section 2.7 can answer “optimal adherence to CPAP.”

Response: In section 2.3.4.1, Good adherence to CPAP at 3 months was defined as a CPAP use for ≥4 h per night on ≥ 70% of nights within a consecutive thirty-day period anytime during the last 90 days of therapy.

7. Page 8, Section 2.4.1, line 5: I suggest spelling out EPT at its first appearance.

Response: Thank you for your suggestion. The abbreviation (TPE) of Therapeutic patient education was spelt out at its first appearance in the background [Over the last two decades, several interventions have been proven to increase CPAP adherence, including technological advances in CPAP devices, augmented clinical support (ACS) (more frequent appointments, telephone calls, home visits), therapeutic patient education (TPE), cognitive behavioral therapy and motivational enhancement. There is an increasing tendency for most clinicians to offer their patients a combination of ACS and TPE].

8. Page 9, Section 2.4.2, lines 1–2: The sentence appeared twice within few lines on this page. I suggest

reducing the redundant words as much as possible.

Response: We apologize for the redundancy and have removed it, as noted in the manuscript. In the paragraph just before section 2.4.4, we referred to the educational team without providing details. Information regarding the educational team was mentioned exclusively in section 2.4.2.

9. Page 9, Section 2.4.5: I suggest calling out Table 1 in Section 2.4.5. The previous calling out in

Section 2.4.1 may not be ideal.

Response: As you suggested, table 1 was called out in Section 2.4.5. instead of section 2.4.1.

10. Page 10, Section 2.6, lines 21–22: This answered Comment 5 above. I suggest revising the earlier

sentence where Comment 5 referred to.

Response: Section 2.6 and Figure 1 have been revised as you suggested. All details are available in response to comment 5.

11. Page 11, Section 2.7: Were there sample size and power calculation to determine the required sample

sizes per group?

Response: We did not calculate the sample size because we consider this a pilot study.

12. Page 11, Section 2.7, line 7: The intervention had two groups. I suggest clarifying what “3 or more

groups” referred to.

Response: The ANOVA test will be used for comparing two variables with three categories each.

13. Table 1: There were only two timestamps, reception and return. I suggest adding a timestamp for each

row. Also, please check the return home timestamp. Was 12 AM correct?

Response: Thank you for this interesting suggestion. As you recommended, we have added a timestamp for each row. We also corrected the return time: it is 12:00 pm and not 12:00 am.

---

## [Decision Letter · Decision Letter 1]

11 Mar 2025

PONE-D-24-46553R1Educational video combined with augmented clinical support to improve CPAP use in patients with obstructive sleep apnea-hypopnea syndrome: a randomized controlled trial protocolPLOS ONE

Dear Dr. Sameh,

Thank you for submitting your manuscript to PLOS ONE. After careful consideration, we feel that it has merit but does not fully meet PLOS ONE’s publication criteria as it currently stands. Therefore, we invite you to submit a revised version of the manuscript that addresses the points raised during the review process.

We look forward to receiving your revised manuscript.

Kind regards,

Yongzhong Guo, Ph.D

Academic Editor

PLOS ONE

Journal Requirements:

Additional Editor Comments:

Authors only make revisions according to reviewer , but are not required to add results as suggested by reviewer 1.

Reviewers' comments:

Reviewer's Responses to Questions

**Comments to the Author**

1. Does the manuscript provide a valid rationale for the proposed study, with clearly identified and justified research questions?

Reviewer #1: Yes

Reviewer #3: Yes

2. Is the protocol technically sound and planned in a manner that will lead to a meaningful outcome and allow testing the stated hypotheses?

Reviewer #1: Yes

Reviewer #3: Yes

3. Is the methodology feasible and described in sufficient detail to allow the work to be replicable?

Reviewer #1: Yes

Reviewer #3: Yes

4. Have the authors described where all data underlying the findings will be made available when the study is complete?

Reviewer #1: Yes

Reviewer #3: No

5. Is the manuscript presented in an intelligible fashion and written in standard English?

Reviewer #1: Yes

Reviewer #3: Yes

6. Review Comments to the Author

You may also provide optional suggestions and comments to authors that they might find helpful in planning their study.

Reviewer #1: While the protocol itself was well written, it would be more meaningful to have the data from the results to comment more about it. I would suggest the manuscript be submitted after the results have been generated.

Reviewer #3: I appreciate the authors’ careful and detailed responses. The responses and the revised manuscript help me

better understand the goal of the manuscript. Here is a list of specific comments to the revised manuscript.

1. Following the response to previous Comment #7, “EPT” was likely a typo of “TPE.” The term EPT

had several appearances in the manuscript. Please confirm the typo and correct it accordingly.

Otherwise, please define EPT.

2. Apologies for any confusion. The response to previous Comment #12 was unclear to me, so I relied

on the authors’ justification.

7. PLOS authors have the option to publish the peer review history of their article (what does this mean? ). If published, this will include your full peer review and any attached files.

**Do you want your identity to be public for this peer review?** For information about this choice, including consent withdrawal, please see our Privacy Policy .

Reviewer #1: No

Reviewer #3: No

---

## [Author Response · Author response to Decision Letter 2]

12 Mar 2025

Dear Editor and Reviewers,

I wish to extend my gratitude for your insightful comments. On behalf of my co-authors, I sincerely appreciate the time and effort you have dedicated to reviewing our manuscript titled "Educational Video Combined with Augmented Clinical Support to Improve CPAP Use in Patients with Obstructive Sleep Apnea-Hypopnea Syndrome: A Randomized Controlled Trial Protocol."

Your feedback has proven invaluable to our work. We have carefully considered your suggestions and revised the manuscript accordingly. Our responses to your comments are outlined in a point-by-point format below. Furthermore, we have submitted a revised version of the manuscript, with the changes highlighted in yellow for your convenience.

We look forward to your evaluation of our revisions and remain open to making any additional modifications that may further enhance the quality of the paper. Thank you once again for your attention and consideration.

Best regards,

Sameh MSAAD, MD

Faculty of Medicine. University of Sfax-Tunisia

Journal requirements

Response: Dear editor, we have rewired all references and we confirmed they are correct and complete.

2. Additional Editor Comments: Authors only make revisions according to reviewer, but are not required to add results as suggested by reviewer 1.

Response: Thank you for clarifying this point. We don’t have any results available.

Reviewers' comments

Reviewer #1

1. While the protocol itself was well written, it would be more meaningful to have the data from the results to comment more about it. I would suggest the manuscript be submitted after the results have been generated.

Response: Dear reviewer, we appreciate your comment. We have no results available.

Reviewer #3

I appreciate the authors’ careful and detailed responses.

The responses and the revised manuscript helped me

better understand the goal of the manuscript. Here is a list of specific comments to the revised manuscript.

Response: Dear reviewer, thank you very much for all your valuable comments and suggestions that have helped us improve our manuscript and make our study protocol clearer and more coherent.

1. Following the response to previous Comment #7, “EPT” was likely a typo of “TPE.” The term EPT

had several appearances in the manuscript. Please confirm the typo and correct it accordingly. Otherwise, please define EPT.

Response: We sincerely apologize for the oversight. The abbreviation TPE, which stands for therapeutic patient education, has been clearly defined, and the typographical error in the manuscript has been corrected (highlighted in yellow).

2. Apologies for any confusion. The response to previous Comment #12 was unclear to me, so I relied

on the authors’ justification.

Response: We regret that our response was not clear enough. As you have mentioned, our study will include only two groups. However, to compare three or more categories across two groups, ANOVA is the recommended statistical test to determine if there are significant differences in means among the categories.

---

## [Editor Report · Decision Letter 2]

19 Mar 2025

Educational video combined with augmented clinical support to improve CPAP use in patients with obstructive sleep apnea-hypopnea syndrome: a randomized controlled trial protocol

PONE-D-24-46553R2

Dear Dr.Sameh,

We’re pleased to inform you that your manuscript has been judged scientifically suitable for publication and will be formally accepted for publication once it meets all outstanding technical requirements.

Kind regards,

Yongzhong Guo, Ph.D

Academic Editor

PLOS ONE
---

## [Editor Report · Acceptance letter]

PONE-D-24-46553R2

PLOS ONE

Dear Dr. Sameh,

I'm pleased to inform you that your manuscript has been deemed suitable for publication in PLOS ONE. Congratulations! Your manuscript is now being handed over to our production team.

Kind regards,

on behalf of

Dr. Yongzhong Guo

Academic Editor

PLOS ONE